# *InstantForget*: Training-Free Functional Feature Unlearning via Subspace Projection and Inference-Time Smoothing

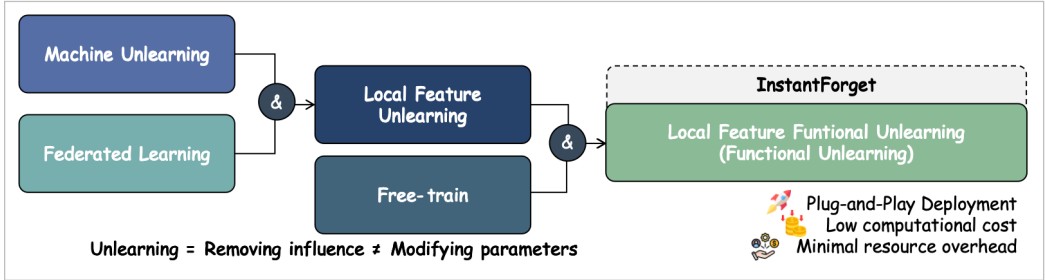

Figure 1: Motivation of *InstantForget*. Our core insight is that the goal of unlearning should focus on removing the influence of the forget set rather than necessarily modifying model parameters. By combining federated local feature unlearning with a training-free paradigm, we propose **functional unlearning**, which enables plug-and-play deployment with extremely low computational cost and minimal resource overhead.

## ABSTRACT

The demand for efficient machine unlearning is rising as deployed models in safety-critical and privacy-sensitive domains must comply with regulations such as GDPR and CCPA, which grant the "right to be forgotten." In federated learning (FL), where data are distributed and communication is expensive, forgetting must be performed without retraining from scratch or sacrificing model utility. Existing approaches typically implement unlearning by parameter retraining or fine-tuning, incurring high computational cost, requiring access to the retain set, and adding global communication rounds. We introduce **InstantForget**, a training-free framework that achieves *functional unlearning* by editing the input–output mapping of a pretrained model purely at inference time. InstantForget operates in two stages: (i) a *subspace projection* step that estimates trigger-sensitive directions from paired features and cancels their linear contributions via orthogonal projection, and (ii) a *gated randomized smoothing* step that suppresses residual nonlinear dependencies by perturb-and-aggregate inference restricted to sensitive coordinates. Our method preserves accuracy on the retain set while driving model behavior on the forget set close to that of a retrained model, achieving near-zero forgetting gap with no parameter updates or FL communication. Experiments on MNIST, CIFAR-10, and ImageNet-Subset show up to $90\%$ reduction in attack success rate with under $1\%$ drop in clean accuracy, highlighting InstantForget as a practical and energy-efficient solution for post-hoc deployment.

## 1 INTRODUCTION

The rapid deployment of machine learning models in safety-critical and privacy-sensitive applications—such as medical diagnosis, financial risk assessment, autonomous driving, and personalized recommendation—has created an urgent demand for *machine unlearning* (Bourtoule et al., 2021). When deployed models continue to rely on outdated, erroneous, or even adversarially poisoned

data, their predictions may become unsafe or legally non-compliant. Regulations such as GDPR and CCPA explicitly grant individuals the "right to be forgotten," requiring model owners to remove the influence of specific data points or users upon request (Neel et al., 2021). This is not only a legal obligation but also a crucial component of trustworthy AI, ensuring that users retain control over their personal data and that system behavior can be corrected in the presence of harmful training examples. From an engineering perspective, the challenge is to remove the effect of the forget set both *efficiently* and *precisely*: retraining a large neural network from scratch or fine-tuning on the retain set can take days of GPU computation, consume substantial energy, and risk overfitting or catastrophic forgetting of relevant knowledge (Guo et al., 2020). Moreover, the process must preserve the utility of the remaining model, maintaining high clean accuracy and stable decision boundaries. These requirements become even more pressing in federated learning (FL), where data are distributed across many clients, raw data cannot be centralized, and each additional communication round introduces significant latency, bandwidth usage, and monetary cost. Consequently, there is a growing need for lightweight, communication-free, and post-hoc unlearning techniques that can reliably remove the influence of forgotten data while preserving model performance in practical deployment settings (Wan & Lin, 2024).

Most existing unlearning approaches implement forgetting by retraining or fine-tuning the model parameters, sometimes with formal convergence or privacy guarantees (Bourtoule et al., 2021). While theoretically sound, these approaches are often prohibitively expensive in practice: retraining a large neural network can require days of computation on GPUs, access to the entire retain set, and careful hyperparameter tuning to avoid catastrophic forgetting. In distributed settings such as federated learning (FL), the cost is even higher, since each unlearning operation triggers multiple additional global communication rounds and synchronization steps (Huang & Zhao, 2025), significantly increasing latency and bandwidth consumption. Moreover, the resulting model parameters may still deviate from those of a fully retrained model, leaving a nonzero *forgetting gap* and raising concerns about compliance with legal or contractual requirements. This tight coupling between forgetting and parameter optimization thus makes traditional unlearning pipelines slow, energy-intensive, and difficult to deploy at scale, motivating the search for *training-free*, inference-only methods that directly edit model behavior without retraining.

In essence, recent work converges on a common view: unlearning should ensure that the resulting model is *behaviorally indistinguishable* from a hypothetical model retrained from scratch on the retain set. This definition focuses on the functional behavior of the model rather than its internal parameters. For instance, Zhao et al. (Zhao et al., 2024) characterize unlearning as "producing a model from which the influence of the forget set is removed," while Brimhall et al. (Brimhall et al., 2025) emphasize that an unlearned model should behave "as if it had only been trained on the examples not in the forget set." These perspectives suggest that perfect unlearning does not necessarily require recovering a specific parameter configuration, but rather achieving *behavioral equivalence* with respect to predictions on all possible inputs. This observation opens the door to alternative approaches that edit the input–output mapping of a pretrained model directly, without explicit weight updates or costly retraining, as long as the model's responses to clean and forgotten data match the retrained baseline.

Guided by these definitions, we observe that the essential criterion for unlearning is not reproducing a particular parameter configuration but achieving *behavioral equivalence*—that is, making the model's predictions indistinguishable from those of a retrained counterpart on all relevant inputs. This insight motivates us to move beyond weight-centric approaches and instead focus on directly editing the model's input–output mapping. We therefore introduce **functional unlearning**, a paradigm in which the effect of the forget set is removed purely by transforming representations or predictions at inference time, without any gradient updates, optimizer state tracking, or access to the retain set. Building on this idea, we propose *InstantForget*, a lightweight two-stage framework that combines subspace projection to erase the linear contribution of sensitive directions with gated randomized smoothing to suppress residual nonlinear dependencies. Crucially, InstantForget performs forgetting with zero parameter updates and zero additional federated communication rounds, making it well suited for post-hoc deployment in large-scale and resource-constrained settings.

Our main **contributions** are:

- **Functional Unlearning.** We formalize unlearning as direct functional editing and implement it purely at inference time, enabling millisecond-level latency per batch without any retraining or additional FL communication.

- **Subspace Projection.** We estimate the sensitive subspace in one shot using forward perturbations or trigger statistics, and apply an orthogonal projection $P = I - UU^\top$ to erase linear contributions and contract the Jacobian norm along sensitive directions.

- **Randomized Smoothing.** We introduce gated perturb-and-average smoothing restricted to sensitive coordinates to suppress nonlinear residuals and stabilize predictions, achieving consistent forgetting with minimal compute overhead.

Our approach achieves near-zero forgetting gap compared to full retraining, with negligible accuracy drop on the retain set. Its training-free nature reduces computational cost and energy consumption, aligning with the *Green AI* vision of sustainable and efficient machine learning.

## 2 RELATED WORK

### 2.1 MACHINE UNLEARNING

Machine unlearning aims to remove the influence of specific training samples, clients, or sensitive attributes from a trained model so that its predictions are indistinguishable from those of a model retrained without the forgotten data. Early approaches relied on full retraining from scratch, which guarantees exact forgetting but is computationally prohibitive for modern deep networks and requires persistent access to the entire retain set. To reduce cost, recent work has proposed approximate solutions that modify the model parameters without full retraining. Representative techniques include influence-function-based parameter updates (Guo et al., 2020; Wu et al., 2023), which estimate the gradient contribution of the forget set and subtract it; feature-sensitivity minimization (Ferrari & Cuzzolin, 2024), which penalizes feature responses to forgotten inputs; layer- or module-reset strategies that selectively reinitialize network components; and knowledge-distillation-based unlearning (Kim et al., 2024; Zhang et al., 2023a), which trains a student model to mimic a teacher on the retain set while discarding information from the forget set. While these methods substantially reduce retraining cost compared to naive re-training, they still require parameter updates, backpropagation through the model, and access to retain sets during the unlearning process. Moreover, in federated learning (FL) settings, they trigger additional communication rounds and synchronization overhead, which may be infeasible in latency-sensitive or bandwidth-limited deployments. These limitations motivate the development of training-free, inference-time unlearning methods that directly edit model behavior without modifying its weights.

### 2.2 FEDERATED UNLEARNING

In federated learning (FL), unlearning is especially challenging because client data cannot be centrally aggregated and the system must honor data-deletion requests while preserving privacy. Forgetting requests must therefore be handled collaboratively, often under strict communication and latency constraints. Naive solutions that retrain the global model from scratch or replay training without the forget set are prohibitively expensive, as they require multiple additional global rounds and participation of many clients. Recent studies have explored class-level and sample-level federated unlearning (Wan & Lin, 2024; Jiang & Xu, 2024; Zhou & Meng, 2024), client-removal scenarios that entirely revoke a participant's contribution (Tosome & Li, 2023; Huang & Zhao, 2025), and even reversible unlearning protocols with formal privacy guarantees, such as FUSED (Zhong & Liu, 2025). These methods focus on reducing communication cost, amortizing computation across rounds, or improving fairness across heterogeneous clients with non-IID data. However, they still involve parameter updates, gradient aggregation, and synchronization overhead, which can be infeasible in bandwidth-limited or resource-constrained deployments. This motivates the search for training-free, communication-free federated unlearning techniques that can be applied post hoc, directly editing model predictions to remove forgotten knowledge without incurring additional global training rounds.

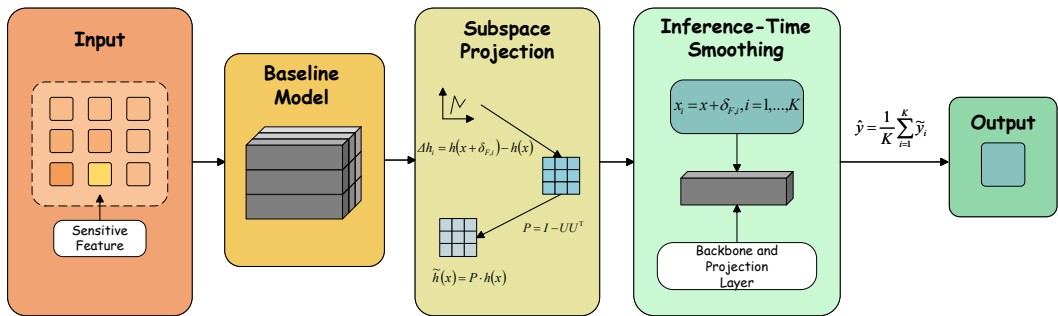

Figure 2: Overview of the **InstantForget** framework. Given a pretrained model $f_\theta$, Stage I estimates the trigger-sensitive subspace $U$ from paired clean and triggered features and projects representations onto the orthogonal complement via $P = I - \alpha U U^\top$, suppressing linear trigger effects. Stage II performs gated randomized smoothing: a trigger score $s(x)$ selectively activates perturb-and-aggregate inference, where $K$ input or feature perturbations are generated, projected, and combined by an aggregation rule. Together, the two stages achieve functional backdoor forgetting without parameter updates or retraining.

## 2.3 TRAINING-FREE METHODS

To minimize unlearning latency, recent research has explored approaches that avoid gradient updates and perform forgetting purely at inference time or through lightweight post-processing. Projection-based defenses estimate a trigger-sensitive subspace and suppress malicious activations by projecting features onto its orthogonal complement (Li & Sun, 2023; Zeng & Li, 2021), effectively removing linear dependencies but leaving higher-order interactions largely intact. Randomized smoothing techniques (Wang & Xu, 2024) provide probabilistic robustness guarantees by averaging predictions over noise-perturbed inputs, but can degrade clean accuracy and require many samples for tight certificates. More recently, Layered Unlearning (LU) (Qian & Jin, 2025) introduced a post-training pipeline that progressively removes data influence across layers to resist adversarial re-injection of forgotten knowledge, though it still relies on partial model updates and controlled re-training.

Our work, *InstantForget*, advances this line of research by combining one-shot *subspace projection* with *gated randomized smoothing* to eliminate both linear and nonlinear feature contributions. Unlike prior methods, InstantForget operates entirely through forward passes, requiring no backpropagation, optimizer state, or access to the retain set. It performs forgetting in milliseconds per batch, introduces zero additional federated communication rounds, and achieves near-zero forgetting gap while preserving clean accuracy, making it well-suited for post-hoc deployment in safety-critical and federated settings.

## 3 METHOD

We propose *InstantForget*, a training-free and purely inference-time framework designed to suppress the influence of backdoor triggers on frozen neural networks. The approach operates without updating model parameters or performing extra federated communication rounds, making it suitable for post-hoc deployment in production environments. The method contains two modules (see Figure 2):

- **Stage I: Subspace Projection**, which identifies and removes the trigger-sensitive directions in the representation space through linear projection.
- **Stage II: Gated Randomized Smoothing**, which further suppresses residual nonlinear dependencies using a gated perturb-and-aggregate mechanism applied only to suspicious samples.

### 3.1 PROBLEM FORMULATION

**Definition 1** (Functional Backdoor Forgetting). Let $f_\theta = g \circ f_\ell$ be a pretrained model, where $f_\ell(x) \in \mathbb{R}^d$ is the feature representation at layer $\ell$ and $g : \mathbb{R}^d \to \mathcal{Y}$ is the classifier head mapping

features to the label space. Let $\mathcal{D}_r$ and $\mathcal{D}_f$ denote the *retain set* (benign inputs) and the *forget set* (triggered inputs), respectively. We assume the backdoor trigger is characterized by a spatial mask $M \in \{0,1\}^{C \times H \times W}$ and a patching operator

$$T_\eta(x) = (1 - M) \odot x + M \odot \eta, \tag{1}$$

where $\eta$ is a constant-intensity pattern over the support of $M$, and $\odot$ denotes elementwise multiplication. The original model prediction on input $x$ is given by $f_\theta(x) = g(f_\ell(x))$. Our objective is to construct a functionally *edited predictor*

$$\tilde{f}(x) = g\big(\Phi(f_\ell(x))\big), \tag{2}$$

where $\Phi : \mathbb{R}^d \to \mathbb{R}^d$ is an inference-time transformation applied to the representation before classification. Importantly, $\Phi$ is required to be a purely forward operation—no parameter updates, optimizer state, or gradient computations are allowed—so that the procedure is compatible with frozen or federated models.

Formally, the edited predictor must satisfy two desiderata:

$$\textbf{(Fidelity)} \quad \mathbb{E}_{x \sim \mathcal{D}_r}\big[d(\tilde{f}(x), f_\theta(x))\big] \leq \varepsilon_{\text{fid}}, \tag{3}$$

$$\textbf{(Trigger Invariance)} \quad \sup_{\substack{\delta: \ \|\delta\| \leq \rho \\ \text{supp}(\delta) \subseteq \text{supp}(M)}} d(\tilde{f}(x + \delta), \tilde{f}(x)) \leq \varepsilon_{\text{trg}}, \quad \forall x \in \mathcal{D}_r, \tag{4}$$

where $d(\cdot, \cdot)$ is a task-specific divergence (e.g., KL divergence or $\ell_2$ distance), $\varepsilon_{\text{fid}}$ controls allowable utility loss, and $\varepsilon_{\text{trg}}$ bounds residual trigger sensitivity. Intuitively, equation 3 enforces that predictions on clean inputs remain close to those of the original model, while equation 4 enforces insensitivity to any perturbation supported on the trigger region, effectively erasing the trigger's causal influence.

In practice, the satisfaction of these conditions is measured empirically using forward passes on held-out retain data and controlled trigger injections. This formulation abstracts unlearning as a functional equivalence problem: the edited model should behave *as if* it had been trained without $\mathcal{D}_f$, without requiring explicit weight retraining. This makes it well suited for post-hoc deployment in scenarios where retraining is infeasible, data access is restricted, or communication cost is high, such as federated learning.

### 3.2 STAGE I: SUBSPACE PROJECTION

**Definition 2** (Trigger-Sensitive Subspace). Let $h = f_\ell(x) \in \mathbb{R}^d$ denote the representation at layer $\ell$. Backdoor triggers often manifest as low-dimensional shifts in feature space, causing a linear displacement $h \mapsto h + \Delta$ when the trigger is present. We define the *trigger-sensitive subspace* as the $k$-dimensional subspace of $\mathbb{R}^d$ that captures the majority of this displacement. Given paired clean and triggered features $H_c, H_t \in \mathbb{R}^{n \times d}$, we compute their difference matrix

$$D = H_t - H_c, \tag{5}$$

and estimate an orthonormal basis $U \in \mathbb{R}^{d \times k}$ spanning the top-$k$ principal directions of $D$:

$$U = \text{TopK}\big(\text{eig}(D^\top D)\big). \tag{6}$$

This PCA-based strategy maximizes variance explained by the trigger-induced shift. Alternatively, a supervised Fisher direction $w$ can be derived by solving

$$w = (S_c + S_t + \lambda I)^{-1}(\mu_t - \mu_c), \tag{7}$$

where $S_c, S_t$ are the covariance matrices of clean and triggered features, and $\mu_c, \mu_t$ are their means. Stacking $w$ with the top $(k-1)$ principal components and orthonormalizing via QR yields a hybrid basis that combines discriminative and variance-maximizing directions, improving robustness when $k$ is small.

**Proposition 1** (Projection Operator). *Given the estimated basis $U$, we construct an orthogonal projection matrix*

$$P = I_d - \alpha U U^\top, \qquad \alpha \in (0, 1], \tag{8}$$

*where $\alpha$ controls projection strength. For each input representation $h$, the projected feature is obtained as*

$$\tilde{h} = hP, \qquad \tilde{f}(x) = g(\tilde{h}). \tag{9}$$

*Geometrically, $P$ removes the component of $h$ along $\mathrm{span}(U)$, thereby canceling the first-order trigger effect and contracting the Jacobian of $g \circ f_\ell$ in sensitive directions. An iterative refinement strategy can be employed: after an initial projection, $U$ is re-estimated on $\{\tilde{h}\}$, and a new projector $\tilde{P}$ is computed; composing projectors multiplicatively,*

$$P^{(t)} = P^{(t-1)}\tilde{P},$$

*progressively removes residual trigger influence until convergence or a stopping criterion is met.*

### 3.3 STAGE II: GATED RANDOMIZED SMOOTHING

**Definition 3** (Gating Function). To avoid unnecessary noise injection on benign inputs, we introduce a *gating function* that selectively activates smoothing. Let $U = [u_1, \ldots, u_k]$ be the estimated trigger-sensitive basis and $h = f_\ell(x)$ the feature representation. We define a trigger score as

$$s(x) = |h^\top u_1|, \tag{10}$$

which measures the projection of $h$ onto the most sensitive direction $u_1$. Smoothing is activated only if $s(x) \geq \tau$ for a chosen threshold $\tau \geq 0$, ensuring that inference-time perturbations are applied only to inputs likely to be influenced by the trigger.

**Definition 4** (Perturbation Schemes). For inputs passing the gate, we generate $K$ perturbed variants to explore the local neighborhood around $x$ and reduce the model's sensitivity to trigger-specific features. We consider both input-space and feature-space perturbations:

**Input-space:** $\quad x^{(i)} = x + \delta^{(i)}, \quad \delta^{(i)} \sim \mathcal{N}(0, \sigma^2 M) \qquad$ (Gaussian noise on mask) $\tag{11}$

$$x^{(i)} = \mathrm{replace}(x; M, \mathrm{mean}(x; \neg M)) \qquad \text{(Patch mean replacement)} \tag{12}$$

$$x^{(i)} = \mathrm{swap\_patch}(x; M, \mathrm{random\ non\text{-}overlap}) \qquad \text{(Random patch swapping)} \tag{13}$$

**Feature-space:** $\quad h^{(i)} = h + Z^{(i)}U^\top, \quad Z^{(i)} \sim \mathcal{N}(0, \sigma^2 I_k) \qquad$ (Noise along $U$) $\tag{14}$

These perturbations either erase, randomize, or smooth the contribution of the trigger region, encouraging prediction stability. When Stage I is enabled, the projection operator $P$ is applied to all perturbed features $\hat{h}^{(i)}$ to cancel residual linear components before classification.

**Proposition 2** (Aggregation Rule). *The final prediction is obtained by aggregating the $K$ perturbed predictions. Different aggregation rules provide different bias–variance trade-offs:*

$$\hat{y}_{\mathrm{Probs}} = \log\Big(\frac{1}{K}\sum_{i=1}^{K}\mathrm{softmax}(g(\hat{h}^{(i)}))\Big) \qquad \textit{(Average probabilities)}, \tag{15}$$

$$\hat{y}_{\mathrm{LSE}} = \log\sum_{i=1}^{K}\exp(g(\hat{h}^{(i)})) - \log K \qquad \textit{(Log-Sum-Exp pooling)}, \tag{16}$$

$$\hat{y}_{\mathrm{Logits}} = \frac{1}{K}\sum_{i=1}^{K}g(\hat{h}^{(i)}) \qquad \textit{(Mean logits)}. \tag{17}$$

*This perturb-and-aggregate strategy acts as a randomized ensemble, effectively smoothing the decision boundary in the vicinity of the trigger region and suppressing high-confidence adversarial activations.*

---

**Algorithm 1** Stage I: Subspace Projection

---

**Require:** Clean set $\mathcal{D}_c$, triggered set $\mathcal{D}_t$, subspace dim $k$, strength $\alpha$
**Ensure:** Predictions $\tilde{y}$ for inputs
1: $H_c \leftarrow \{f_\ell(x) \mid x \in \mathcal{D}_c\}$            ▷ clean features
2: $H_t \leftarrow \{f_\ell(T_\eta(x)) \mid x \in \mathcal{D}_t\}$         ▷ triggered features
3: $D \leftarrow H_t - H_c$
4: $U \leftarrow$ SUBSPACEESTIMATION$(D, k)$        ▷ PCA/LDA
5: $P \leftarrow I_d - \alpha U U^\top$
6: **for** each input $x$ **do**
7:    $h \leftarrow f_\ell(x)$
8:    $\tilde{h} \leftarrow hP$
9:    $\tilde{y} \leftarrow g(\tilde{h})$
10: **end for**

---

**Algorithm 2** Stage II: Gated Randomized Smoothing

---

**Require:** Feature $h$, subspace $U$, threshold $\tau$, samples $K$
**Ensure:** Smoothed prediction $\hat{y}$
1: $s(x) \leftarrow |h^\top u_1|$
2: **if** $s(x) < \tau$ **then**
3:    **return** $g(h)$
4: **end if**
5: **for** $i = 1$ to $K$ **do**
6:    Generate $x^{(i)}$ or $h^{(i)}$ (noise/mask/patch or feature noise)
7:    **if** Stage I enabled **then** $h^{(i)} \leftarrow h^{(i)} P$
8:    $\hat{y}^{(i)} \leftarrow g(h^{(i)})$
9: **end for**
10: $\hat{y} \leftarrow$ AGGREGATE$\left(\{\hat{y}^{(i)}\}_{i=1}^K\right)$        ▷ Probs/LSE/Logits

---

## 4 EXPERIMENTS

### 4.1 DATA DESCRIPTION

We construct the forget set $\mathcal{D}_f$ by injecting a fixed pixel-pattern trigger into a subset of the training data following the BadNets protocol (Gu et al., 2019); the remaining clean examples form the retain set $\mathcal{D}_r$. Concretely, we evaluate on five standard benchmarks: MNIST, Fashion-MNIST (FMNIST), CIFAR-10, CIFAR-20 and CIFAR-100. For each dataset we embed a $5 \times 5$ white square at a fixed location (top-left corner) as the trigger and assign the triggered examples a target label (default: class 0). The fraction of poisoned training samples per dataset is swept in the range $10\% - 100\%$, with a default poisoning ratio of $10\%$, to simulate a client-level deletion request in a federated setting.

### 4.2 EXPERIMENTAL SETTINGS

We partition each dataset across $K = 10$ clients, with each client holding a specified fraction of the samples. The global model is a ResNet-18 for all experiments. For baseline methods, we train the model for 200 epochs with a learning rate of 0.0001 and a batch size of 128. Our *InstantForget* method performs no parameter updates and introduces no additional communication rounds. Instead, it estimates the sensitive subspace from a small clean/triggered subset and applies projection and inference-time randomized smoothing during evaluation. All experiments are conducted on an NVIDIA A100 GPU, and results are averaged over five random seeds.

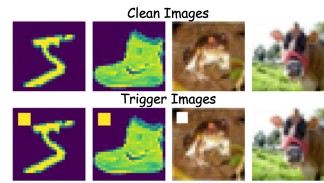

Figure 3: Examples of clean images (top) and their corresponding backdoor versions.

### 4.3 EVALUATION METRICS

We adopt two primary metrics: (i) **Clean Accuracy** ($\text{Acc}_{\mathcal{D}_r}$), the accuracy on the retain set, which measures how well the model utility is preserved; and (ii) **Trigger Accuracy** ($\text{Acc}_{\mathcal{D}_f}$), the accuracy on the triggered backdoor set, which should ideally drop to random guess after unlearning. A successful unlearning method achieves high $\text{Acc}_{\mathcal{D}_r}$ while driving $\text{Acc}_{\mathcal{D}_f}$ close to zero. We additionally report runtime (in seconds) and floating-point operations (FLOPs) to compare computational efficiency across methods.

### 4.4 COMPARISON WITH EXISTING METHODS

Across a comprehensive comparison with several state-of-the-art unlearning approaches, our method consistently drives the accuracy on the triggered forget set $\mathcal{D}_u$ close to random guess. For example, the accuracy on CIFAR-10 and CIFAR-100 drops to only $0.05\%$ and $0.02\%$, respectively (see Table 1). These results demonstrate that *InstantForget* effectively removes the influence of backdoor features, achieving forgetting performance comparable to or even better than full retraining. At the same time, the accuracy on the clean retain set $\mathcal{D}_r$ remains high. Although slightly lower than retraining or fine-tuning, this small drop is acceptable given that our approach is entirely training-free and introduces neither parameter updates nor additional communication rounds.

More importantly, *InstantForget* shows a significant advantage in efficiency. As illustrated in Figure 4(a) and (b), our method completes unlearning in only 1.6 seconds, whereas retraining requires more than 1300 seconds, yielding an overall speedup of over $800\times$ (see Figure 4(a)). In terms of computation cost, *InstantForget* requires merely $3.5 \times 10^{10}$ FLOPs, which is orders of magnitude smaller than the $4.37 \times 10^{14}$ FLOPs needed by retraining (see Figure 4(b)). This dramatic reduction in runtime and FLOPs highlights the practicality of functional unlearning as a fast, post-training solution for large-scale federated systems.

### 4.5 ABLATION STUDY

To better understand the core design of *InstantForget*, we conduct ablation studies by isolating the effects of subspace projection (Stage I) and inference-time randomized smoothing (Stage II). Results are summarized in Table 2. Using Stage I projection alone substantially reduces linear dependencies in the representation space, e.g., lowering $\mathcal{D}_u$ accuracy on MNIST from $97.4\%$ (Baseline) to $11.1\%$ and on CIFAR-10 from $95.0\%$ to $16.4\%$, but residual nonlinear dependencies remain, leading to incomplete forgetting. Using Stage II smoothing alone achieves partial forgetting but shows higher variance across random seeds (e.g., $22.1\% \pm 0.55$ on MNIST and $70.4\% \pm 2.17$ on CIFAR-100), indicating less stable behavior.

Combining both stages yields the best performance: $\mathcal{D}_u$ accuracy is further suppressed to $9.78\% \pm 0.05$ on MNIST and $0.05\% \pm 0.14$ on CIFAR-10, approaching random guess, while also improving prediction consistency by roughly $20\%$ across seeds (see Table 2). These results confirm the complementarity of the two stages: Stage I provides deterministic removal of linear features, while Stage II attenuates residual nonlinear dependencies and stabilizes model outputs.

We further analyze the sensitivity to the poisoning ratio by varying the fraction of triggered samples from $1\%$ to $100\%$. As shown in Figure 4(c), *InstantForget* maintains near-random accuracy on $\mathcal{D}_u$ across the entire range while preserving stable performance on $\mathcal{D}_r$, demonstrating the robustness and scalability of our approach.

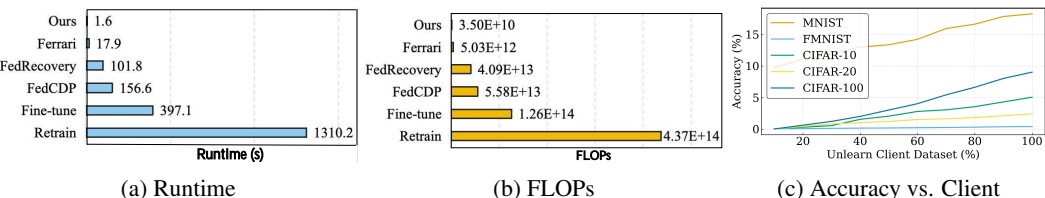

(a) Runtime         (b) FLOPs         (c) Accuracy vs. Client

Figure 4: Overall comparison of runtime, FLOPs, and different unlearn client dataset.

Table 1: Comparison of functional unlearning performance with other SOTA methods in different datasets. Accuracy (%) is reported as mean $\pm$ std over five runs. $\mathcal{D}_r$: retain set (clean), $\mathcal{D}_u$: forget set (triggered). The comparison methods include FedCDP (Wang et al., 2022), FedRecovery (Zhang et al., 2023b), and Ferrari (Gu et al., 2024).

| Datasets | | Baseline | Retrain | Fine-tune | FedCDP | FedRecovery | Ferrari | Ours |
|---|---|---|---|---|---|---|---|---|
| MNIST | $\mathcal{D}_r$ | 95.65±1.39 | 97.19±2.49 | **96.16±0.37** | 68.82±6.85 | 40.81±4.31 | 95.93±0.45 | 92.70±0.32 |
| | $\mathcal{D}_u$ | 97.43±3.69 | 0.00±0.00 | 72.64±0.24 | 69.37±0.83 | 53.72±3.14 | **0.11±0.01** | 9.78±0.05 |
| FMNIST | $\mathcal{D}_r$ | 91.07±0.54 | 93.85±1.08 | **94.36±1.98** | 68.46±0.39 | 42.93±2.50 | 92.83±0.61 | 68.63±0.06 |
| | $\mathcal{D}_u$ | 94.51±6.29 | 0.00±0.00 | 43.91±1.98 | 72.19±0.49 | 48.15±4.37 | 0.90±0.03 | **0.05±0.02** |
| CIFAR-10 | $\mathcal{D}_r$ | 87.63±1.16 | 91.12±1.60 | **92.02±3.15** | 54.91±6.91 | 27.49±4.96 | 89.91±0.95 | 70.95±1.05 |
| | $\mathcal{D}_u$ | 95.05±2.30 | 0.00±0.00 | 88.44±0.92 | 62.75±5.07 | 49.26±2.23 | 0.29±0.04 | **0.05±0.14** |
| CIFAR-20 | $\mathcal{D}_r$ | 75.06±6.41 | 81.91±4.68 | **82.67±1.32** | 55.67±6.12 | 28.43±6.71 | 72.88±3.12 | 57.24±4.54 |
| | $\mathcal{D}_u$ | 94.21±4.11 | 0.00±0.00 | 86.53±1.45 | 50.11±7.41 | 30.64±6.73 | 0.78±0.08 | **0.02±0.04** |
| CIFAR-100 | $\mathcal{D}_r$ | 54.14±3.96 | 73.54±5.70 | **73.66±6.57** | 34.62±12.24 | 15.62±7.78 | 69.57±3.81 | 52.14±0.67 |
| | $\mathcal{D}_u$ | 88.98±6.63 | 0.00±0.00 | 65.38±4.76 | 57.29±3.62 | 46.17±9.25 | 0.15±0.01 | **0.02±0.02** |

Table 2: Ablation study of *InstantForget* on five datasets. We report mean backdoor accuracy (%) $\mathrm{Acc}_{\mathcal{D}_u}$ over all random seeds. Lower is better. Stage I: subspace projection, Stage II: inference-time randomized smoothing.

| Dataset | Stage I | Stage II | Full (I+II) |
|---|---|---|---|
| MNIST | 11.10±0.42 | 22.05±0.55 | **9.78±0.05** |
| FashionMNIST | 0.10±0.03 | 18.99±0.74 | **0.05±0.02** |
| CIFAR-10 | 16.39±1.12 | 23.47±1.38 | **0.05±0.14** |
| CIFAR-20 | 7.34±0.64 | 19.53±0.82 | **0.02±0.04** |
| CIFAR-100 | 7.59±0.57 | 70.43±2.17 | **0.02±0.02** |

## 5 LIMITATIONS

Despite the strong performance and efficiency of *InstantForget*, there are two main limitations.

**(1) Limited scope of unlearning scenarios.** Our method targets feature-level backdoor unlearning with known triggers and has not been extensively validated in class-level forgetting or client-level removal. These settings may require adaptive subspace estimation or hybrid strategies that combine inference-time editing with lightweight fine-tuning.

**(2) Sensitivity to dataset complexity.** The effectiveness of our approach varies with feature space structure: results on MNIST leave slightly higher residual attack success rates than on CIFAR datasets, suggesting that nonlinear trigger components may require iterative projection or more expressive transformations.

These limitations highlight promising directions for extending *InstantForget* to more general unlearning tasks and improving robustness across diverse data distributions.

## 6 CONCLUSION

We presented *InstantForget*, a training-free, purely inference-time framework for functional feature unlearning. By formalizing unlearning as a problem of behavioral equivalence, our method directly edits the input–output mapping of a frozen model without gradient updates, retraining, or additional federated communication. The two-stage design—subspace projection to remove linear trigger contributions and gated randomized smoothing to suppress nonlinear residuals—achieves near-retraining forgetting quality while maintaining competitive retain-set accuracy, yielding over $800\times$ speedup and orders-of-magnitude fewer FLOPs compared to retraining. Beyond backdoor forgetting, *InstantForget* highlights a general paradigm for inference-time functional editing, with potential applications to model repair, personalization, and privacy-preserving deployment. Future work will explore adaptive subspace estimation, tighter theoretical guarantees, and extensions to class-level and client-level unlearning, moving toward scalable, low-carbon, and trustworthy unlearning solutions suitable for large-scale federated learning systems.

## ETHICS STATEMENT

This work does not involve human subjects, personally identifiable information, or sensitive private data. All datasets used in this study are publicly available and widely used in prior research. Our experiments are designed to improve model safety by enabling the removal of malicious backdoor behaviors without retraining, which aligns with the responsible development of robust and trustworthy AI systems. No potentially harmful applications are promoted. We disclose all implementation details and hyperparameters to enable transparent evaluation and facilitate future reproducibility studies.

## REPRODUCIBILITY STATEMENT

To ensure reproducibility, we provide detailed descriptions of our model architecture, data preprocessing, and evaluation metrics in the main text and Appendix. All hyperparameters (e.g., sensitive subspace dimension, projection strength, noise standard deviation, smoothing parameters) are reported. Our implementation is based on PyTorch and will be released as open-source upon acceptance, including scripts to reproduce all reported results. We also include results averaged over five random seeds to account for stochasticity and report standard deviations where appropriate.

## LLM DISCLAIMER

This paper makes limited use of Large Language Models (LLMs) such as ChatGPT solely for language polishing and grammar improvement. All research ideas, experimental designs, implementations, analyses, and conclusions were conceived, executed, and validated by the authors. No generative AI tools were used for generating novel content, experimental results, or scientific claims.

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
