# OpenReview forum: "InstantForget: Training-Free Functional Feature Unlearning via Subspace Projection and Inference-Time Smoothing"
_ICLR.cc/2026/Conference — ICLR 2026 Conference Withdrawn Submission_

### Official Review · Reviewer_dyrj · 2025-10-26

**Soundness:** 1
**Presentation:** 2
**Contribution:** 2
**Rating:** 4
**Confidence:** 4

**Summary:**

This paper introduces InstantForget, a novel training-free framework for machine unlearning. Its core innovation lies in reframing unlearning as a problem of achieving behavioral equivalence rather than parameter equivalence. Instead of retraining or fine-tuning the model, InstantForget performs functional unlearning by directly editing the model's input-output mapping during inference. This is achieved through a two-stage process: Stage I employs subspace projection to eliminate linear contributions of trigger-sensitive features, and Stage II uses gated randomized smoothing to suppress residual nonlinear dependencies.

**Strengths:**

1. The method eliminates the need for backpropagation or parameter updates, relying solely on forward passes. This results in a dramatic speedup and a reduction in FLOPs compared to retraining.

2. It operates locally without introducing additional communication rounds, making it ideal for bandwidth-constrained or latency-sensitive federated learning scenarios.

3. By editing feature representations at inference time rather than modifying the model's weights, it offers a plug-and-play unlearning solution that preserves the original model parameters.

**Weaknesses:**

1. The method is primarily validated on backdoor trigger unlearning with known trigger patterns. Its effectiveness remains unproven for more general unlearning scenarios, such as class-level or client-level forgetting, and it is likely inapplicable in scenarios with unknown triggers.

2. The higher residual attack success rate on simpler datasets like MNIST suggests limited capacity for suppressing highly nonlinear triggers. Furthermore, its performance on high-dimensional, complex data is not sufficiently validated.

3. The effectiveness of the projection mechanism is highly dependent on the accurate estimation of the trigger-sensitive subspace. Inaccurate estimation could lead to either over-forgetting or under-forgetting.

4. As shown in Table 1, the proposed method suffers a significant drop in accuracy on the retain set compared to methods like Ferrari, which maintains high clean accuracy. This substantial utility loss raises concerns about the practical effectiveness of the unlearning process.

5. Given the considerable accuracy penalty and the fact that other methods (e.g., Ferrari) may offer comparable computational efficiency in terms of runtime/FLOPs (as suggested in Figures 4a and 4b) while achieving far superior accuracy, the overall benefit and necessity of a training-free approach are called into question.

**Questions:**

1. Given the substantial performance gap on the retain set between InstantForget and parameter-update methods like Ferrari, how can the trade-off between efficiency and model utility be justified for practical applications?

2. Could the framework be extended to handle class-level or client-level unlearning? What architectural or methodological changes would be required?

3. How sensitive is the method to the dimensionality (k) of the estimated sensitive subspace? Is there a risk of overfitting to the small set of paired samples used for subspace estimation?

4. The method assumes prior knowledge of the trigger pattern. How could it be adapted to a more challenging scenario where the trigger is unknown or has to be inferred?

5. Figures 4a and 4b suggest that the Ferrari method has computational costs comparable to InstantForget. Does this comparison include the cost of the fine-tuning process for Ferrari, or only the unlearning application time?

---

### Official Review · Reviewer_PQqH · 2025-10-30

**Soundness:** 2
**Presentation:** 3
**Contribution:** 1
**Rating:** 2
**Confidence:** 5

**Summary:**

This paper introduces a training-free subspace projection at the feature level, where the original features are projected onto a direction that is orthogonal to the feature space of the forgotten data while remaining as close as possible to the regular data space for forward output. Robustness is further enhanced through multiple noise perturbations and averaging, ultimately achieving the goal of machine unlearning. The method is evaluated on small-scale datasets on MNIST and CIFAR.

**Strengths:**

1. Considering machine unlearning from a training-free perspective is an inspiring direction, as it can help significantly reduce the computational cost for tasks where retraining is particularly expensive.

2. The paper’s method section is clearly explained and well organized, making it easy to follow and understand.

**Weaknesses:**

1. The approach proposed in this paper aims to achieve machine unlearning without modifying the original parameters by inserting simple modules. This design is fundamentally flawed because the original parameters remain fully preserved, making the model highly vulnerable to attacks. The data intended to be forgotten can easily be recovered by disabling the inserted module, meaning the method cannot truly protect data privacy.

2. Aside from the design concept, the algorithmic implementation cost remains uncontrollable. Although the method does not require training, it still needs to compute the feature differences between the retain data and the forget data. While the forget set may be relatively small, the retain set is typically unknown. In fact, for modern large language models, the amount of data that needs to be retained is enormous, making the computation of feature representations entirely infeasible. Therefore, this approach is difficult to extend to larger-scale datasets and models.

3. The paper lacks sufficient theoretical analysis. It relies mainly on heuristic discussions to justify the feasibility of the proposed approach, without providing rigorous theoretical support to ensure how the inserted projection module affects convergence and generalization.

4. The experimental scale in the paper is relatively small, making it difficult to determine whether the proposed method can effectively generalize and transfer to larger-scale models.

5. The performance of this approach appears to be quite poor in the experiments. The results in Table 1 show a significant degradation on the original dataset—more than 10% lower than other baselines—which is unacceptable.

**Questions:**

1. How can authors protect the proposed method from not being attack by backdoor injection?

2. Can the authors evaluate the privacy strength of this approach?

3. Can the authors provide a more thorough analysis to demonstrate that this operation does not compromise the original results in terms of convergence and generalization performance, or that any potential impact remains within an acceptable range?

4. Although the performance on the forgetting dataset in Table 1 is very poor, this seems to occur because all datasets perform poorly overall—the retained-data performance is also extremely low. Is this a general phenomenon? Could it be that the introduced projection module introduces excessive error, leading to the degradation across all datasets?

---

### Official Review · Reviewer_KyBL · 2025-10-30

**Soundness:** 3
**Presentation:** 3
**Contribution:** 2
**Rating:** 4
**Confidence:** 4

**Summary:**

- InstantForget is an inference-time method that requires no retraining or fine-tuning and leaves the model parameters unchanged.

- The core idea is to identify trigger-sensitive subspaces from the unlearning set, orthogonally project features onto the complement of those subspaces before the classifier head, and use gated randomized smoothing to apply the projection only to inputs likely influenced by the trigger.

- InstantForget is evaluated comprehensively through multi-dataset benchmarks, utility vs. forgetting trade-off, efficiency metrics, and component-wise ablation, demonstrating strong performance and practicality for training-free unlearning.

**Strengths:**

- The method operates only at inference time and does not add training-time computational overhead.
- Results on color-shifted datasets are superior to baselines.
- The use of an orthogonal basis is interesting.

**Weaknesses:**

- However, Table 1 suggests unstable performance: (i) overall, the method is not better than Ferrari when considering both ACC on retaining and unlearning sets; (ii) it significantly compromises clean accuracy, especially on the FMNIST dataset, and often shows 15–25% lower retention accuracy than Ferrari.
- The requested knowledge remains in the model since it is never actually unlearned; please discuss why this is acceptable in an unlearning context. Can the insights about orthogonal bases be extended to truly remove knowledge from the model?
- The method targets trigger unlearning only. How would it handle more practical settings such as unlearning an identity, a class, or a physical attribute (e.g., color, glasses) rather than an artificial BadNet-style trigger?
- Overall, while the method is interesting, the contribution feels somewhat limited.

**Questions:**

1. How sensitive is the unlearning effectiveness to the chosen subspace dimensionality $k$? Have you analyzed how over- or under-estimating $k$ affects both forgetting and retention accuracy?

2. The method assumes access to known triggers in the unlearning set. How does InstantForget perform when the trigger distribution or location differs slightly from those seen during subspace estimation?

3. Can the subspace projection and smoothing framework generalize to unlearning other types of information (e.g., class-level, identity, or attribute-level forgetting)? If not, what are the main barriers?

4. Equations (3) and (4) formalize fidelity and trigger invariance objectives. How are the hyperparameters $\varepsilon_{\text{fid}}$ and $\varepsilon_{\text{trg}}$, or the projection strength $\alpha$, chosen in practice to balance these two goals?

5. What aspects of the feature space or model architecture contribute most to the variation against dataset complexity, and does the total number of classes affect the performance?

---

### Note · Authors · 2025-11-12

I have read and agree with the venue's withdrawal policy on behalf of myself and my co-authors.